# Superconducting diode effect via conformal-mapped nanoholes

Yang-Yang Lyu[1,2,8], Ji Jiang[3,4,8], Yong-Lei Wang [1✉], Zhi-Li Xiao [2,5✉], Sining Dong[1], Qing-Hu Chen[3], Milorad V. Milošević [4], Huabing Wang[1,6], Ralu Divan[7], John E. Pearson[2], Peiheng Wu[1,6], Francois M. Peeters[4] & Wai-Kwong Kwok[2]

A superconducting diode is an electronic device that conducts supercurrent and exhibits zero resistance primarily for one direction of applied current. Such a dissipationless diode is a desirable unit for constructing electronic circuits with ultralow power consumption. However, realizing a superconducting diode is fundamentally and technologically challenging, as it usually requires a material structure without a centre of inversion, which is scarce among superconducting materials. Here, we demonstrate a superconducting diode achieved in a conventional superconducting film patterned with a conformal array of nanoscale holes, which breaks the spatial inversion symmetry. We showcase the superconducting diode effect through switchable and reversible rectification signals, which can be three orders of magnitude larger than that from a flux-quantum diode. The introduction of conformal potential landscapes for creating a superconducting diode is thereby proven as a convenient, tunable, yet vastly advantageous tool for superconducting electronics. This could be readily applicable to any superconducting materials, including cuprates and iron-based superconductors that have higher transition temperatures and are desirable in device applications.

[1] Research Institute of Superconductor Electronics, School of Electronic Science and Engineering, Nanjing University, Nanjing, China. [2] Materials Science Division, Argonne National Laboratory, Argonne, IL, USA. [3] Department of Physics, Zhejiang University, Hangzhou, China. [4] NANOlab Center of Excellence and Department of Physics, University of Antwerp, Antwerp, Belgium. [5] Department of Physics, Northern Illinois University, DeKalb, IL, USA. [6] Purple Mountain Laboratories, Nanjing, China. [7] Center for Nanoscale Materials, Argonne National Laboratory, Argonne, IL, USA. [8] These authors contributed equally: Yang-Yang Lyu, Ji Jiang. ✉email: yongleiwang@nju.edu.cn; xiao@anl.gov

Semiconductor diodes made from p–n junctions have low resistances in one current direction and high resistances in the other. They are the essential units in modern electronics, widely used in computation, communication, and sensing[1]. Due to the finite resistance of a semiconductor, energy loss in a semiconductor diode is inevitable. Therefore, a superconducting diode with dissipationless current remains highly desired for electronic devices with ultralow power consumption. However, the realization of an ideal superconducting diode is scarce and highly anticipated[2]. Only recently, a notable superconducting diode was realized in an artificially fabricated noncentrosymmetric superlattice, composed of alternating epitaxial films of niobium, vanadium, and tantalum[3]. So far, the superconducting diode effect of nonreciprocal superconducting-to-normal transition in an ordinary superconducting material has not been demonstrated. Here we introduce a simple method to achieve the superconducting diode with uni-directional supercurrent in a conventional superconducting film through nanoengineering.

A primary function of a diode is rectification, a process that converts an alternating current (AC) into a direct current (DC). Recently, nonreciprocal charge transport with inequivalent effects from forward and backward currents was observed in superconducting samples with interfacial symmetry breaking, e.g., in $Bi_2Te_3$/FeTe heterostructure[4] and gate-induced polar superconductor $SrTiO_3$[5]. However, the rectification ratios for these two-dimensional superconductors were very small. To produce large enough rectification signals, a polar film with alternating layers of three different materials (containing superconductor and nonsuperconductors) was fabricated to directly realize the superconducting diode effect[3]. In all the above material systems, the broken inversion symmetries are all perpendicular to the sample plane, and a precisely aligned in-plane magnetic field perpendicular to the applied current is required to break the time-reversal symmetry. On the other hand, a broken spatial inversion symmetry in the sample plane of a noncentrosymmetric

superconductor such as the two-dimensional $MoS_2$[6,7] can also induce nonreciprocal charge transport. In this case, an out-of-plane magnetic field is applied for inducing the nonreciprocal charge transport. This suggests the potential for generating directional supercurrent by breaking the in-plane spatial inversion symmetry. However, similar to polar films with out-of-plane broken inversion symmetry[4,5], the rectification ratio in the two-dimensional superconducting $MoS_2$ was not sufficient to directly demonstrate the superconducting diode effect of nonreciprocal superconducting-to-normal transition. Here, we achieved a clear superconducting diode effect in a superconducting film with broken inversion symmetry in the sample plane, using artificially patterned conformal arrays of nanoscale holes in the film (Fig. 1). This method could be conveniently applied to a variety of commonly available superconductors, including high transition temperature (high-$T_c$) superconductors.

## Results

**Giant rectification effect.** Our device was fabricated on a film of amorphous superconducting alloy $Mo_{0.79}Ge_{0.21}$ (MoGe) without intrinsic anisotropies. The film thickness is 50 nm and was patterned into a 50-μm-wide microbridge containing two sections, as shown in Fig. 1a. One section contains two repetitive conformal mapped triangular arrays of nanoscale holes (Fig. 1b), while the other section contains a regular triangular array of nanoscale holes as a reference (Supplementary Fig. 1a). A conformal array is a two-dimensional structure created through a conformal (angle-preserving) transformation to a regular lattice[8,9]. It preserves the local ordering of the regular lattice but induces a gradient distribution which breaks the in-plane inversion symmetry. The nominal hole diameter is 110 nm (inset of Fig. 1b). A detailed sample fabrication process can be found in Methods. The superconducting transition temperature of the sample at zero external magnetic field was approximately 6.0 K (Supplementary Fig. 2).

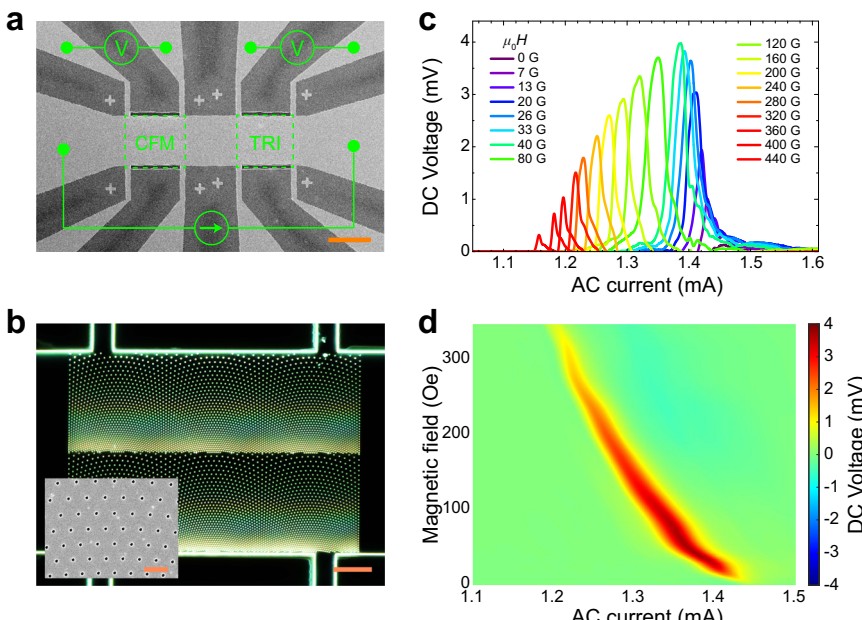

**Fig. 1 Giant rectification effect. a** Scanning electron microscopy image of a superconducting MoGe microbridge containing two sections with conformal mapped triangular (CFM) and regular triangular (TRI) arrays of nano-holes, respectively. The AC current is applied horizontally, and the DC voltage is measured with the top or the bottom leads. The magnetic field is applied perpendicularly to the sample plane. Scale bar, 40 μm. **b** Dark field optical image of the CFM section with conformal arrays of nano-holes. Scale bar, 10 μm. The inset is a SEM image of the nano-holes with diameter of 110 nm. Scale bar of inset, 1 μm. **c**, **d** Voltage curves and color maps of the AC current dependence of the DC voltage at various magnetic fields measured at the temperature of 5.8 K from the CFM section.

The magnetic fields are applied out-of-plane. We demonstrate the superconducting diode effect by directly applying an AC current (30 kHz) and measuring the DC voltage as displayed in Fig. 1c, d. Clear rectification signals of net DC voltages are observed at certain AC currents and magnetic fields for the section with conformal nanopatterns. In contrast, the rectification signal for the section with a uniform triangular array of holes is negligible at all currents and magnetic fields (Supplementary Fig. 1b). This highlights the critical role of the broken spatial inversion symmetry in realizing the superconducting diode effects. Additionally, with increasing magnetic field, the AC current threshold responsible for producing the rectification voltage shifts to lower values. At zero magnetic field there is almost no rectification voltage. In contrast, the rectification signal gradually increases to a maximum value with increasing magnetic field, and then decreases with further increase of the magnetic field.

It is well known that in Type-II superconductors, the magnetic field penetrates the sample in the form of quantized magnetic fluxes, each carrying exactly one flux-quantum $\Phi_0 = 2.1 \times 10^{-15}$ Wb. The maximum rectification signal occurs at a magnetic field of ~40 Oe (Fig. 1c), corresponding to the matching field at which the density of flux-quanta is equivalent to the average density of the perforations. The hole diameter of 110 nm allows one flux-quantum trapped by each hole[9]. We also measured a sample containing a conformal array of holes with the same distribution but with a hole diameter of 220 nm (Supplementary Fig. 3a), which allows for two flux-quanta to be trapped by each hole[9]. In that case the maximum rectification occurs around 80 Oe (Supplementary Fig. 3b), i.e., exactly twice of that for the sample with 110 nm holes. Such a magnetic field dependence of the rectification implies that the motion of flux-quanta plays a crucial role in the observed rectification signal.

Previous experimental and theoretical studies widely showed that rectification signals in superconductors can be generated from a flux-quanta diode effect, i.e., the directional (or ratchet) motion of flux-quanta. For example, rectification was induced in superconducting thin films containing artificial nano-defects with asymmetric geometries[10–23] and in heterostructures containing asymmetrically shaped magnetic nanostructures on top of superconductors[24–30]. Besides introducing local asymmetries, breaking global inversion symmetries is another way to induce ratchet motion of superconducting vortices. The latter approach includes introducing inhomogeneously distributed nano-defects with a long-range gradient density[31–33] as well as patterning a uniform superconductor into an asymmetric shape[34–37]. Additionally, apart from spatial asymmetries, time-asymmetric currents can also generate a flux-quantum diode effect[38,39]. Recent computer simulations proposed a ratchet flux-quantum motion with dynamic potentials[40]. More specifically, a flux-quantum diode effect was predicted by molecular dynamics simulations in a model system containing conformal pinning landscapes[31], similar to the sample configuration of this work. However, the most significant result of our data in Fig. 1c, d is that the amplitude of the rectification voltage is at the level of millivolts, which is three orders of magnitude larger than that induced by the flux-quantum diode effect, for example the microvolts rectification voltage recently realized in a microbridge fabricated using the same material and the same dimensions as in this work[24]. This implies that the giant rectification observed here should be different from the nominal flux-quantum diode effect.

**Insight from simulations**. To prove the observed rectification originates from the superconducting diode effect with non-reciprocal superconducting-to-normal transition, we carried out the time-dependent Ginzburg–Landau (G–L) simulations (see Methods). The simulated rectification (Fig. 2a) shows excellent consistency with the experimental data in Fig. 1c. The top panel

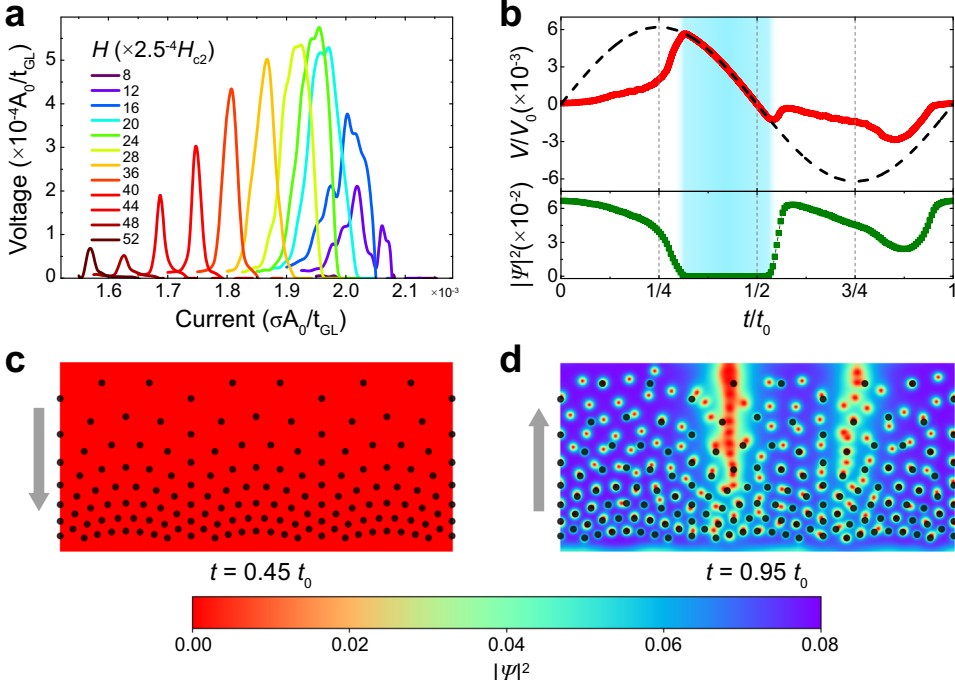

**Fig. 2 Ginzburg–Landau simulations of the superconducting diode effect. a** Simulated AC current dependence of the rectification DC voltage at various magnetic fields. **b** Calculated time-dependent voltage (top panel) and order parameter (bottom panel) over a period of a sine wave AC current. Unit $t_0$ is the period of the sine wave. **c, d** Color maps of the superconducting Cooper-pair density (screenshots of Supplementary Movie 1) at 0.45 and 0.95 of the sine-wave period, respectively. Black dots indicate nanoholes. The red spots in **d** show flux-carrying flux-quanta. The gray arrows indicate the direction of the driven flux-quanta.

of Fig. 2b shows the simulated voltage signal over a full period of a sine-wave AC current. In the blue-shaded region in Fig. 2b the voltage curve follows exactly the sine wave, indicating a linear voltage–current relationship in this region. This means that the sample is in a constant-resistance state within this region. The bottom panel of Fig. 2b shows clearly that in the same blue-shaded region, the sample is in the normal state (superconducting order parameter is zero). Figure 2b also shows that a current-driven superconducting-to-normal transition emerges at the first half-period of the sine-wave current with positive polarity, while there is no such transition in the second half-period with negative polarity. Supplementary Movie 1 and Supplementary Fig. 4 show the evolution of the superconducting Cooper-pair density, clearly revealing the nonreciprocal superconducting-to-normal transition. Figure 2c, d shows screenshots of Supplementary Movie 1 in the positive and negative current regimes, respectively, demonstrating the normal state with zero-order parameter and the superconducting state with flux-flow driven by the supercurrent. These simulation results clearly demonstrate that the observed rectification originates from the superconducting diode effect rather than the flux-quantum diode effect. Since the voltage in the normal state is much larger than that in the flux-flow state, the rectification signal from the superconducting diode effect is much larger than that arising solely from the flux-quantum motion.

**Physical mechanism**. Our simulations suggest that the superconducting diode effect originates from nonequivalent nucleation and evolution of hot spots (Supplementary Movie 2 and Supplementary Fig. 5). As demonstrated in Supplementary Movie 3 and Supplementary Fig. 6, the flux-quanta acquire a gradient distribution when they are driven into motion. The density of the flux-quanta in the frontline (at bottom in Supplementary Movie 3 and Supplementary Fig. 6 for positive current) is lower than that in the back (at top in Supplementary Movie 3 and Supplementary Fig. 6 for positive current). When the density gradient of the moving flux-quanta matches that of the conformal hole array, the flux pinning is more effective. This leads to a slower flux-flow in one direction. Subsequently, the faster moving flux-quanta under a positive current generate higher energy dissipation, leading to larger Joule heating than under a negative current. The larger Joule heating drives the system from the superconducting state into the normal state at smaller currents. That is, there is a range of currents in which the film is driven to the normal state at positive currents while remaining in the superconducting state at negative currents, thereby resulting in the superconducting diode effect. Furthermore, flux-quanta residing in the nanoscale holes minimize the energy of the superconducting condensate, hence flux-quanta are effectively attracted by holes. As shown in our simulated flux-quantum trajectories in Supplementary Fig. 7, closely spaced nanoholes form guiding channels for flux-quanta, highlighted by the dotted fans in Supplementary Fig. 7a, b, thus enabling strong flux-quantum funneling effect for the flux-quanta in the front under positive current and the flux-quanta in the back under negative current. Supplementary Movie 2 and Supplementary Fig. 5 show that hot spots are always induced by the flux-quanta in the front (at the bottom) when $I > 0$ (Supplementary Fig. 5b) and on top when $I < 0$ (Supplementary Fig. 5j), owing to the fact that the flux-quanta in the front move faster and hence produce larger heating. When $I > 0$, the flux-quanta in the front locate at the fan areas (Supplementary Fig. 7a) of the conformal lattice, where strong flux funneling concentrates the moving flux-quanta. Therefore, the hot spots nucleate easily in the fan areas of the conformal lattice under positive current. In contrast, when $I < 0$, the faster moving flux-quanta in the front

locate on the opposite side to the fan areas, and the nucleation of hot spots in the absence of strong flux funneling requires a much higher applied current. Thus, our superconducting diode effect with nonequivalent nucleation and evolution of hot spots originates from nonreciprocal dynamic density matching (Supplementary Movie 3 and Supplementary Fig. 6) and funneling effects (Supplementary Fig. 7) of moving flux as it encounters the asymmetric conformal array of perforations.

**DC and quasi-DC probes**. To demonstrate the superconducting diode effect in a more direct way, we conducted experiments of DC voltage versus DC current. The result clearly shows different critical currents for positive ($+I$) and negative ($-I$) applied currents (Fig. 3a), demonstrating the nonreciprocal current-induced superconducting-to-normal transitions. In order to directly and quantitatively compare our observed giant rectification effect with the superconducting diode effect, we designed further experiments using quasi-DC waveforms of the applied current: a fully-rectified standard sine wave (Abs-AC) and a half-rectified sine wave (Semi-AC), as shown in the insets of Fig. 3b. These quasi-DC currents oscillate like a sine wave AC current, but with fixed positive or negative polarities. The experimental results of the quasi-DC experiments are shown in Fig. 3b. The microbridge with the conformal array of holes shows a distinct shift in the transition currents (marked by black arrows) with application of positive and negative quasi-DC currents (green curves for Abs-AC and red curves for Semi-AC), while the microbridge with the uniform triangular lattice of holes shows symmetric inverted transitions between quasi-DC currents of different polarities (gray curves). This result clearly demonstrates the superconducting diode effect. By adding the two curves of the quasi-DC currents with positive and negative polarities, we obtain the calculated rectification signals, as shown in Fig. 3c. It shows that the current interval that exhibits significant rectification signals from both quasi-DC experiments (green curve for Abs-AC and red curve for Semi-AC) match perfectly with that obtained from direct AC experiments (black curve). Moreover, the amplitude of the calculated rectification from Semi-AC current experiments is also consistent with that from direct AC experiments, confirming that addition of the positive and negative Semi-AC waveform is equivalent to the standard AC waveform. On the other hand, the addition of the positive and negative Abs-AC waveform corresponds to two standard AC waveforms, and hence the signal strength calculated from the Abs-AC experiments is nearly twice that of the direct AC experiment. We measured the magnetic field dependence of the quasi-DC experiments using Abs-AC currents (Supplementary Fig. 8). In Fig. 3d we plot the current and magnetic field dependences of the rectification signals calculated from the quasi-DC experiments. The results are also consistent with those of direct AC experiments shown in Fig. 1d. Thus, we unambiguously confirm that the observed giant superconducting rectification originates from the superconducting diode effect.

**Tunable superconducting diode**. Since our superconducting diode effect originates from nonequivalent Joule heating induced by flux-flow, our superconducting diode retains all the known properties of flux-quantum diodes, yet vastly surpasses them in tunability. For example, as shown in the inset of Fig. 4a, the diode effect can be easily switched on/off by tuning the magnetic field. The rectification polarity can also be conveniently reversed by flipping the direction of the magnetic field, as displayed by the negative magnetic field dependence of rectification in Fig. 4a. Furthermore, the rectification signal can also be scaled with temperature, which affects the critical current of the superconducting-to-normal transition. Figure 4b

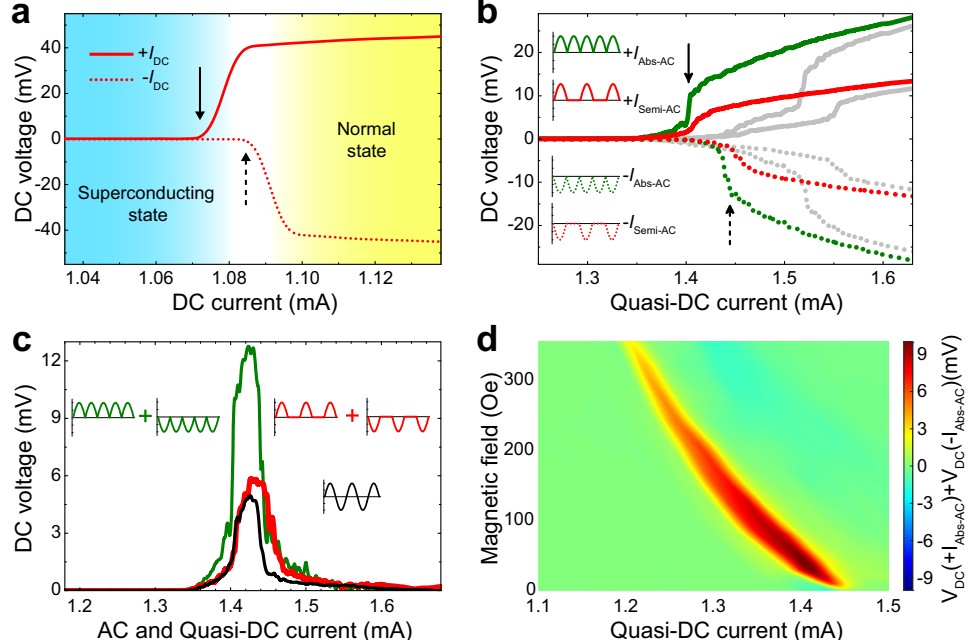

**Fig. 3 DC and quasi-DC probes of the superconducting diode effect. a** DC voltage versus DC currents for positive (solid line) and negative (dashed line) currents at 40 Oe. **b** Quasi-DC experiments at 40 Oe. Green and red curves are measured using Abs-AC and Semi-AC currents (insets), respectively. Gray curves are corresponding measurements of the reference section TRI. **c** Comparison of the calculated rectification signals from Quasi-DC measurements (green, for Abs-AC currents; red, for Semi-AC currents) and those from direct AC experiments (black). **d** Color map of the magnetic field and current-dependent calculated rectification signals using results from Supplementary Fig. 8. The experiments were all conducted at temperature of 5.8 K.

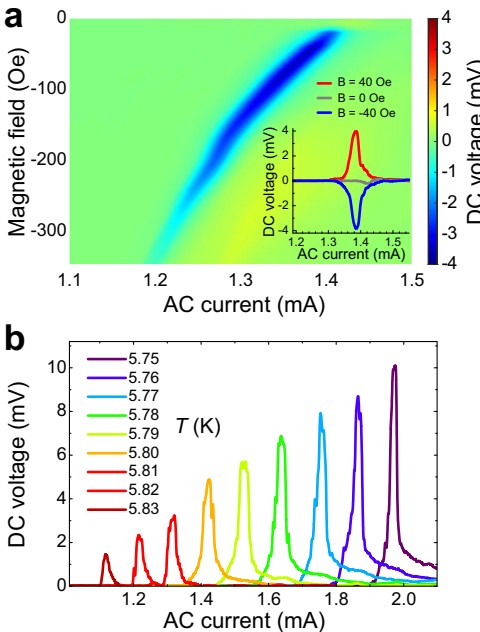

**Fig. 4 Tunable superconducting diode effect and scalable rectification. a** Color maps of the magnetic field and AC current-dependent rectification voltages measured under negative magnetic fields with reversed polarities of flux-quanta (corresponding to the reverse of Fig. 1d). The inset demonstrates the switching between on, off, and reverse polarity of the rectification. **b** Rectification amplitude at various temperatures in a magnetic field of 40 Oe.

shows an enhancement of the rectification when lowering the temperature, directly demonstrating a temperature scalable superconducting diode, presenting yet another advantage, in addition to the giant rectification, of our superconducting diode.

## Discussion

Since the superconducting diode effect arises from nonreciprocal heating effects, optimizing the thermal connections to the environment (e.g., via choice of substrates and/or adjustments in cryogenic cooling) may modify the rectification characteristics, providing additional design opportunities. Furthermore, tuning the frequency of the driving AC current could affect the motion of flux-quanta[18] by modifying the time scales for thermal relaxation, thus providing another knob to control the superconducting diode effect. Tuning the normal-state resistance could be yet another way to modulate the rectification strength, by e.g., adjusting the sample dimensions of length, width and/or thickness, as well as by selecting superconducting materials of different normal-state resistivities. In addition to reversing the magnetic field, inverting the orientation of the conformal array pattern can also reverse the polarity of the superconducting diode. Finally, the demonstrated method of breaking inversion symmetry in the film plane through nanoengineering could be employed on other Type-II superconductors, e.g., high-$T_c$ cuprates and iron-based superconductors, for higher working temperatures and significantly larger operational magnetic fields. This type of scalable superconducting diode would therefore greatly enrich the flexibility of designing advanced dissipationless superconducting electronic devices, with an additional outlook towards rectification effect at radio frequencies[14] for use as ultrasensitive filters and/or receivers for microwave applications and superconducting quantum computing.

## Methods

**Sample fabrication**. Our samples were fabricated on a 50 nm thick $Mo_{0.79}Ge_{0.21}$ (MoGe) superconducting thin film. A 50-µm-wide MoGe microbridge on silicon substrates with an oxide layer was fabricated using the standard lift-off techniques of magnetron sputter and photolithography[9]. The nanoscale holes were created using electron beam lithography followed by reactive ion etching[9]. We fabricated samples with hole diameters of 110 nm and 220 nm for different samples. These hole sizes correspond to the maximum flux-quantum trapping number of one and two, respectively[9]. The microbridge contains two sections (Fig. 1a) of conformally

mapped (Fig. 1b) and triangularly distributed (Supplementary Fig. 1) holes, respectively. The superconducting transition temperature was approximately 6.0 K (Supplementary Fig. 2).

The conformal pattern was created from a triangular lattice with a lattice constant $a = 500$ nm with site coordinates $(x, y)$ in an area of $r_{in} < \sqrt{x^2 + y^2} < r_{out}$ ($r_{in} = 7.9$ μm, $r_{out} = 22.5$ μm). Then the triangular lattice coordinates $(x, y)$ were converted to conformal lattice coordinates $(x', y')$ using the following formula:

$$x' = \begin{cases} r_{out}\left[\arctan\left(\frac{y}{x}\right) + \pi\right] & \text{if } y \leq 0 \\ r_{out}\arctan\left(\frac{y}{x}\right) & \text{if } y > 0 \end{cases}$$

$$y' = \frac{1}{2}r_{out}\ln\left(\frac{r_{out}^2}{x^2 + y^2}\right)$$

The created conformal pattern has the same average site density with a triangular pattern with a lattice constant of 777 nm.

**Experiments**. The transport experiments were carried out using a standard four-probe method. The standard sine wave AC current, the Quasi-DC current (programmed wave function), and the DC current were generated using a Keithley 6221 current source. The DC voltage was measured using a Keithley 2182 A voltmeter. The sample was placed in a superconducting magnet, with the magnetic field applied perpendicular to the sample plane. The measurements were taken at various fixed temperatures with a stability within ±1 mK.

**Time-dependent Ginzburg–Landau (G–L) simulations**. The simulation was conducted on a 2-D rectangular superconducting sample on a heat reservoir holder. The dimensionless time-dependent G–L equations are used to directly observe the flux-quanta behavior in presence of driving current[41–43]:

$$\frac{\partial \psi}{\partial t} = (\nabla - i\mathbf{A})^2\psi + \alpha(1 - T)\psi - |\psi|^2\psi + \chi(\mathbf{r}, t) \tag{1}$$

$$\sigma\frac{\partial \mathbf{A}}{\partial t} = Im\left[\psi^*(\nabla - i\mathbf{A})\psi\right] - \kappa^2\nabla \times \nabla \times \mathbf{A} \tag{2}$$

where $\psi$ is the superconducting order parameter, $\mathbf{A}$ is the vector potential of the magnetic field, $\sigma$ is the conductivity in the normal state, and $\chi(\mathbf{r}, t)$ is a random function used to mimic the quantum fluctuations[43,44]. All lengths are in units of coherence length at zero temperature $\xi(0)$ and time is scaled to $t_{GL} = 4\pi\lambda(0)^2\sigma/c^2$, where $\lambda(0)$ is the penetration length at zero temperature and $c$ is the speed of light. Temperature is scaled by the critical temperature $T_c$, magnetic field is in units of $H_{c2}(0) = \Phi_0/2\pi\xi(0)$, where $\Phi_0$ is the flux quantum, and current density is in units of $j_0 = \sigma\hbar/2e\xi(0)t_{GL}$. In simulations, pinning sites are introduced through the parameter $\alpha$ in Eq.(1) using the so-called $\delta T$ pinning, where the local critical temperature is reduced within the radius of the defects[45,46].

Heat transfer equation is used to describe the effect of Joule heating[43,47]:

$$\nu\frac{\partial T}{\partial t} = \zeta\nabla^2 T + \left(\sigma\frac{\partial \mathbf{A}}{\partial t}\right)^2 - \eta(T - T_0), \tag{3}$$

where $\nu, \zeta, \eta$ are the heat capacity, heat conductivity of the sample, and heat coupling to the holder, respectively. Here we used $\nu = 0.03$, $\zeta = 0.06$, $\eta = 2.0 \times 10^{-4}$ which correspond to intermediate heat removal at our simulation temperature[47]. Equations(1–3) are solved self-consistently using the Crank–Nicholson method[41].

The simulated sample is $L_x = 640\xi(0)$ long and $L_y = 340\xi(0)$ wide. Neumann boundary conditions are used at $y = 0, L_y$ and periodic boundary conditions are applied at $x = 0, L_x$. The magnetic field is applied along the $+z$ direction perpendicular to the superconducting film. External current is introduced through the following boundary conditions for the magnetic field: $H|_{y=0} = H_{ext} - H_I$ and $H|_{y=Ly} = H_{ext} + H_I$, where $H_{ext}$ is the applied magnetic field and $H_I$ is the field induced by the applied current. Note that this approach is only valid for thin superconductors or superconductors which are isotropic in the $z$ direction. The AC current is applied with a fixed period $P = 10^4 t_{GL}$ which is sufficiently long for the vortices to travel across the entire sample within a half-period.

## Data availability
The data that support the findings of this study are available from the corresponding author upon request.

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

## Acknowledgements

This work is supported by the National Key R&D Program of China (2018YFA0209002 and 2017YFA0303002), the National Natural Science Foundation of China (61771235, 61971464, 61727805, 11961141002, 11834005, and 11674285), Jiangsu Excellent Young Scholar program (BK20200008), and Jiangsu Shuangchuang program. Z.L.X., J.E.P., and W.K.K. acknowledge support from the U.S. Department of Energy, Office of Science, Basic Energy Sciences, Materials Sciences and Engineering. Z.L.X. also acknowledges support from the National Science Foundation under Grant No. DMR-1901843. R.D. acknowledges support from the US Department of Energy, Office of Science, Office of Basic Energy Sciences, under contract number DE-AC02-06CH11357. M.V.M. and F.M. P. acknowledge support by the Research Foundation-Flanders (FWO).

## Author contributions

Y.L.W., Z.L.X., and W.K.K. conceived and supervised the project. Y.Y.L., Y.L.W., Z.L.X., and W.K.K. designed the experiments. J.J., Q.C., M.V.M., and F.M.P. designed the simulations and provided theoretical interpretation. Y.Y.L., Y.L.W., S.D., H.W., R.D., and J.E.P. fabricated the samples. Y.Y.L. and Y.L.W. conducted the transport measurements. J.J. conducted the simulations. Y.Y.L., J.J., Y.L.W., M.V.M., and Z.L.X. analyzed the data. Y.Y.L., J.J., Y.L.W., Z.L.X., P.W., M.V.M., and W.K.K. co-wrote the manuscript. All authors contributed to the discussion of the data and the final scientific statement of the manuscript.

## Competing interests

The authors declare no competing interests.
