## [Peer Review File · Nature Communications]

REVIEWER COMMENTS

Reviewer #1 (Remarks to the Author):

This is a very good paper which reports a superconducting diode effect achieved in a conventional superconducting MoGe film patterned with a conformal array of nanoscale holes, which breaks the spatial inversion symmetry. The demonstrated superconducting diode effect is three orders of magnitude larger than that from a flux-quantum diode reported before.

The paper is well written and the experimental result is original and clear. I recommend the publication of this paper in Nature Communications after revision.

My concern is the title of this paper "Programmable superconducting diode". I cannot understand the meaning of "Programmable" here. I think that people imagine such kind of devices like FPGA in semiconductor from the word "Programmable". I recommend the authors to change the title to more specific one like "Superconducting diode effect in MoGe film patterned with a conformal array of nanoscale holes", and to remove the word programmable from the text.

Reviewer #2 (Remarks to the Author):

It is a timely and original contribution on a currently hot topic. The whole set of experimental data supported by theory and simulations appears solid and convincing.

I believe that this manuscript should be considered for publications in Nature Communications, provided that the points outlined below are properly taken into account.

a) The physics of the problem is discussed more in the supplementary material rather in the text, and this weakens the whole manuscript. The section "Microscopic mechanism of the superconducting diode effect" should be somehow incorporated in the text replacing some parts of the manuscript with less relevant information.

b) This can be the basis to better explain why the observed rectification should be different from the mere flux-quantum diode effect. Videos based on Ginzburg-Landau are very powerful, but physical interplaying effects should be better discussed in the text. It is not straightforward the correlation between physical conclusions in lines 130-138 and the videos. Values of physical parameters obtained through simulations are not discussed enough.

c) Have the Authors fabricated samples where the effect is not observed by changing one of the key parameters used to calibrate the whole system?

d) In Fig. 4b the Authors report the dc voltage vs the AC current as a function of the Temperature. Which is the uncertainty they have on the temperature?

Reviewer #3 (Remarks to the Author):

In this manuscript, the authors reports on the operation of a programmable superconducting diode-like device, which exploits an original concept relying on the breaking of inversion symmetry in conformal pinning arrays. They first demonstrate a rectification effect, i. e. the transformation of an ac signal into a dc one, whose amplitude is several orders magnitude larger than in the more conventional superconducting diodes whose operation is based on the flux-quantum diode effect. The results are supported by numerical simulations of the time-dependent Ginzburg-Landau equations. In

particular, videos available as supplementary material illustrate very clearly the operation of the device. The authors also measured directly the diode effect in dc transport experiments in agreement with the rectification effect. Overall, the manuscript presents a very original work, based on a robust set of experiment data, which are well supported by a numerical modeling. Before I can make a final recommendation on publication, I'd like the authors to address the following comments :

- The use of a conformal triangular array of pinning hole being at the heart of the operation of the device, the manuscript should include a brief discussion on the exact design of the array.

- In this work, the authors used an amorphous MoGe alloy superconducting thin films, which is not a standard superconducting material. What is the motivation for this choice ? What about the intrinsic distribution of defects that could also behave as pinning centers for vortices?

- The authors indicate that they do not observe any rectification effect on the regular (i.e. not conformal) triangular array. Is it only because such array does not break the inversion symmetry or is it also related to the pinning properties of the array ? In principle, one expects the triangular array to be more efficient to pin vortices.

- For sake of comparison, I suggest the authors to also show numerical simulation of the time dependent G-L equations for the regular triangular lattice.

- In figure 2b, the authors show the calculated time dependent voltage generated by rectification over a period of a sine wave ac current at 30 kHz. In principle, such quantity should not be too difficult to measure experimentally. Have the authors tried to do such measurements and compared them to the simulation ?

- In my opinion, a discussion on the frequency dependence of the response of the device is really missing in the article. If it cannot be done experimentally, I would appreciate to have some inputs on the dynamics of the system from the time-dependent G-L simulations.

Responses to comments and recommendations of Reviewer #1

Reviewer recommendation

This is a very good paper which reports a superconducting diode effect achieved in a conventional superconducting MoGe film patterned with a conformal array of nanoscale holes, which breaks the spatial inversion symmetry. The demonstrated superconducting diode effect is three orders of magnitude larger than that from a flux-quantum diode reported before. The paper is well written and the experimental result is original and clear. I recommend the publication of this paper in Nature Communications after revision.

Our response

We are grateful to the reviewer for the praise of our work and recommendation towards publication in Nature Communications.

Reviewer comment#1

My concern is the title of this paper “Programmable superconducting diode”. I cannot understand the meaning of “Programmable” here. I think that people imagine such kind of devices like FPGA in semiconductor from the word “Programmable”. I recommend the authors to change the title to more specific one like “Superconducting diode effect in MoGe film patterned with a conformal array of nanoscale holes”, and to remove the word programmable from the text.

Our response

We used the term “programmable” since the diode effect presented here can be tuned with magnetic field and/or temperature. However, we understand the intent behind the reviewer’s comment. We therefore changed the title from “Programmable superconducting diode” to “Superconducting diode effect via conformal-mapped nanoholes”, to address the issue yet retain the appeal of the title.

Responses to comments and recommendations of Reviewer #2

Reviewer recommendation

It is a timely and original contribution on a currently hot topic. The whole set of experimental data supported by theory and simulations appears solid and convincing. I believe that this manuscript should be considered for publications in Nature Communications, provided that the points outlined below are properly taken into account.

Our response

We are grateful to the reviewer for complimenting our work and for recommending its publication in Nature Communications. We followed her/his suggestions on improving the manuscript’s readability and revised it accordingly.

Reviewer comment#1

a) The physics of the problem is discussed more in the supplementary material rather in the text, and this weakens the whole manuscript. The section “Microscopic mechanism of the

superconducting diode effect” should be somehow incorporated in the text replacing some parts of the manuscript with less relevant information.

Our response

Following the Reviewer’s suggestion, we moved the discussion regarding the microscopic mechanism of the superconducting diode effect from the supplementary material section to the main body of the manuscript.

Reviewer comment#2

b) This can be the basis to better explain why the observed rectification should be different from the mere flux-quantum diode effect. Videos based on Ginzburg-Landau are very powerful, but physical interplaying effects should be better discussed in the text. It is not straightforward the correlation between physical conclusions in lines 130-138 and the videos. Values of physical parameters obtained through simulations are not discussed enough.

Our response

In the revised manuscript, we added the discussion of the physical mechanism behind the diode effect, in which we detailed the relevant phenomena as visible in the videos.

Reviewer comment#3

c) Have the Authors fabricated samples where the effect is not observed by changing one of the key parameters used to calibrate the whole system?

Our response

We measured a reference sample with a triangular array of nanoscale holes, which has no broken inversion symmetry. In this case, the superconducting diode effect is absent, as demonstrated in Supplementary Fig. 1.

Reviewer comment#4

d) In Fig. 4b the Authors report the dc voltage vs the AC current as a function of the Temperature. Which is the uncertainty they have on the temperature?

Our response

The superconducting critical current becomes higher when the temperature is decreased. The dc voltage vs the AC current curves were taken at different fixed temperatures with a stability within ± 1 mK. We added this information in the Methods section.

Responses to comments and recommendations of Reviewer #3

Reviewer recommendation

In this manuscript, the authors report on the operation of a programmable superconducting diode-like device, which exploits an original concept relying on the breaking of inversion symmetry in conformal pinning arrays. They first demonstrate a rectification effect, i. e. the transformation of an ac signal into a dc one, whose amplitude is several orders magnitude larger than in the more conventional superconducting diodes whose operation is based on the

flux-quantum diode effect. The results are supported by numerical simulations of the time-dependent Ginzburg-Landau equations. In particular, videos available as supplementary material illustrate very clearly the operation of the device. The authors also measured directly the diode effect in dc transport experiments in agreement with the rectification effect. Overall, the manuscript presents a very original work, based on a robust set of experiment data, which are well supported by a numerical modeling. Before I can make a final recommendation on publication, I'd like the authors to address the following comments:

Our response

We are delighted that the Reviewer finds our work to be “a very original work, based on a robust set of experiment data, which are well supported by a numerical modeling”. We thank the reviewer for this praise and other thoughtful comments.

Reviewer comment#1

- The use of a conformal triangular array of pinning hole being at the heart of the operation of the device, the manuscript should include a brief discussion on the exact design of the array.

Our response

We added the following description and a new reference [Ref.8] on the design of the conformal array in the revised manuscript.

“A conformal array is a two-dimensional structure created through a conformal (angle-preserving) transformation to a regular lattice^{8,9}. It preserves the local ordering of the regular lattice but induces a gradient distribution that breaks the in-plane inversion symmetry.”

Reviewer comment#2

- In this work, the authors used an amorphous MoGe alloy superconducting thin films, which is not a standard superconducting material. What is the motivation for this choice? What about the intrinsic distribution of defects that could also behave as pinning centers for vortices?

Our response

Amorphous MoGe can form very thin homogenous films and has been a platform for studying superconductor-insulator transitions [please see Phys. Rev. Lett. 74, 3037 (1995); Phys. Rev. Lett. 126, 077001 (2021)]. It has been widely used in vortex matter research for decades [Please see Phys. Rev. Lett. 122, 047001 (2019); Nat. Commun. 9, 4922 (2018); Phys. Rev. Lett. 70, 670 (1993)]. Due to its extremely *weak intrinsic pinning* of vortices, MoGe has been a popular choice of material to explore *artificially induced vortex pinning* [please see our previous publications Nat. Nanotechnol. 13, 560 (2018); Phys. Rev. Lett. 111, 067001 (2013)]. With intrinsic pinning absent, the effects from such externally introduced alterations can be clearly uncovered.

Reviewer comment#3

-The authors indicate that they do not observe any rectification effect on the regular (i.e. not conformal) triangular array. Is it only because such array does not break the inversion symmetry or is it also related to the pinning properties of the array? In principle, one expects the triangular array to be more efficient to pin vortices.

Our response

Triangular and conformal arrays have very similar pinning strengths: a triangular array performs slightly better than a conformal array below the first matching field while the latter can do better at higher fields [please see our previous work in Ref. 9]. A triangular array does not induce the diode effect simply because it has no broken inversion symmetry.

Reviewer comment#4

- For sake of comparison, I suggest the authors to also show numerical simulation of the time dependent G-L equations for the regular triangular lattice.

Our response

We conducted G-L simulations for a sample with a regular triangular lattice of holes. The simulated vortex behavior is consistent with the experimental results and is shown in Supplementary Fig. 1c.

Reviewer comment#5

- In figure 2b, the authors show the calculated time dependent voltage generated by rectification over a period of a sine wave ac current at 30 kHz. In principle, such quantity should not be too difficult to measure experimentally. Have the authors tried to do such measurements and compared them to the simulation?

Our response

We did not carry out the time-dependent voltage measurements. Although it is doable, measurements to get time-dependent AC response in Fig. 2b require fast sampling of voltages with microvolt resolutions. Such a data acquisition device is currently not available in our laboratory. On the other hand, our DC and quasi-DC measurements (Fig. 3) provide clear evidence of the nonreciprocal superconducting-to-normal transition.

Reviewer comment#6

- In my opinion, a discussion on the frequency dependence of the response of the device is really missing in the article. If it cannot be done experimentally, I would appreciate to have some inputs on the dynamics of the system from the time-dependent G-L simulations.

Our response

Tuning the frequency of the driving AC current could affect the diode effect by modifying the time scales for thermal relaxation, which may indeed provide another knob to control the superconducting diode.

We did investigate the frequency dependence on the superconducting diode effect. The results are added as Supplementary Fig. S9: the superconducting diode effect has negligible frequency dependence up to 50 kHz, the upper limit of our measurement system. Theoretically, we do expect a suppression of the superconducting diode effect at very high frequencies, where the system does not have enough time to remove the Joule heating when the current reverses its

direction. Measurements at very high frequencies can be complicated, because the signal can be influenced by numerous thermal factors arising from the substrate, the thermal characteristics of the sample, as well as signal dissipation from the connected circuits and wires.

REVIEWERS' COMMENTS

Reviewer #2 (Remarks to the Author):

The manuscript has been properly revised according to the Referees' comments and, in my opinion, can be now published.

Reviewer #3 (Remarks to the Author):

The authors have addressed all the comments and questions of the three reviewers in a convincing way. The manuscript has been significantly improved and I am happy to recommend publication.